# Coarse-to-Fine Dual Encoders are Better Frame Identification Learners

**Kaikai An[1,2]\*, Ce Zheng[1]\*, Bofei Gao[1], Haozhe Zhao[1,2], Baobao Chang[1]†**

[1] National Key Laboratory for Multimedia Information Processing,
School of Computer Science, Peking University
[2] School of Software & Microelectronics, Peking University
{ankaikai,zce1112zslx,gaobofei,hanszhao}@stu.pku.edu.cn
chbb@pku.edu.cn

## Abstract

Frame identification aims to find semantic frames associated with target words in a sentence. Recent researches measure the similarity or matching score between targets and candidate frames by modeling frame definitions. However, they either lack sufficient representation learning of the definitions or face challenges in efficiently selecting the most suitable frame from over 1000 candidate frames. Moreover, commonly used lexicon filtering (*lf*) to obtain candidate frames for the target may ignore out-of-vocabulary targets and cause inadequate frame modeling. In this paper, we propose COFFTEA, a Coarse-to-Fine Frame and Target Encoders Architecture. With contrastive learning and dual encoders, COFFTEA efficiently and effectively models the alignment between frames and targets. By employing a coarse-to-fine curriculum learning procedure, COFFTEA gradually learns to differentiate frames with varying degrees of similarity. Experimental results demonstrate that COFFTEA outperforms previous models by 0.93 overall scores and 1.53 R@1 without *lf*. Further analysis suggests that COFFTEA can better model the relationships between frame and frame, as well as target and target. The code for our approach is available at https://github.com/pkunlp-icler/COFFTEA.

## 1 Introduction

Frame Identification (Gildea and Jurafsky, 2000; Baker et al., 2007) aims to identify a semantic frame for the given target word in a sentence. The frames defined in FrameNet (Baker et al., 1998) are associated with word meanings as encyclopedia knowledge (Fillmore et al., 1976), which represent different events, situations, relations and objects. Identifying frames can contribute to extracting frame semantic structures from a sentence, and

*Equal contribution
†Corresponding author

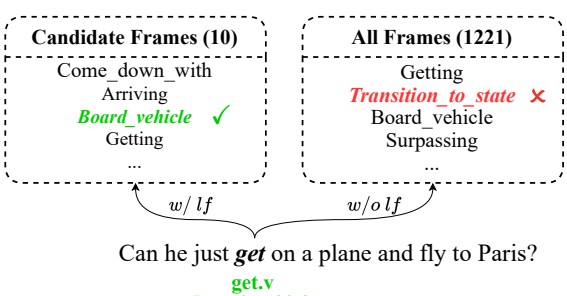

Figure 1: An example of Frame Identification. The target word *get* in the sentence is associated with frame ***Board_vehicle***. **Lexicon filtering** (*lf*) considers frames that can be evoked by **get.v** as candidates.

frame semantic structures can be utilized in downstream tasks, e.g., Information Extraction (Si and Roberts, 2018), Machine Reading Comprehension (Guo et al., 2020).

Previous researches in frame identification selects an appropriate frame for a target by modeling the similarity or matching score between the target and frames. Recent efforts are made to represent frames with definitions in FrameNet (Jiang and Riloff, 2021; Su et al., 2021; Tamburini, 2022; Zheng et al., 2023). However, limited by the large scale of frames, these methods either struggle to effectively model definitions due to the use of a frozen frame encoder or cannot efficiently select appropriate frames from all frames due to the use of a single encoder architecture.

Fig. 1 illustrates an example of frame identification. For the target word "get" in the sentence, we need to select its corresponding frame from over a thousand frames in FrameNet (**All**): ***Board_vehicle*** (A Traveller boards a Vehicle). Previous frame identification methods (Peng et al., 2018; Jiang and Riloff, 2021) typically apply lexicon filtering (**lf**) by selecting the frame from a set of candidate frames (**Candidates**) for the verb "get" (e.g., *Arriving*, *Getting*, *Board_vehicle*, etc.). *lf* effec-

tively improves the accuracy of frame identification. However, the manually defined lexicons is not possible to cover all the corresponding targets for frame, which hinders the extension of frame semantics to out-of-vocabulary target words.

Furthermore, we observed that training *w/o lf* (without *lf*) poses challenges in distinguishing fine-grained differences among candidate frames. Conversely, training *w/ lf* (with *lf*) can lead to an increased distance between the gold frame and other candidates, making it difficult to distinguish the gold frame from irrelevant ones. Consequently, in this paper, our aim is to enhance our model's performance in both the *w/ lf* and *w/o lf* scenarios.

In this paper, we propose COFFTEA, a **Co**arse-to-**F**ine **F**rame and **T**arget **E**ncoder **A**rchitecture. By employing a dual encoder structure, we can effectively store the representations of all frames during inference, enabling us to efficiently compute the similarity between the target and all frames. Furthermore, considering the constraints imposed by the number of frames, we incorporate contrastive objectives to enhance the trainability of the frame encoder, consequently improving our ability to model definitions more accurately. Additionally, employing a distinct frame or target encoder allows us to map certain frames and targets to a vector space and utilize distance measures to depict their relationship.

To address the trade-off between training *w/ lf* and *w/o lf*, we propose a two-stage learning procedure, referred to as coarse-to-fine learning. In the first stage, we employ **in-batch learning**, using the gold frames of other instances within a batch as hard negative examples. This step helps the model learn to discern frames with distinct semantics. In the second stage, we adopt **in-candidate learning**, where we utilize other frames from the candidate set as negative samples. This enables the model to further refine its ability to recognize fine-grained semantic differences. With coarse-to-fine learning, COFFTEA achieves a comprehensive performance in both scenarios of *w/ lf* and *w/o lf*.

Experiments show that COFFTEA, compared to previous methods on two FrameNet datasets, achieves competitive accuracy *w/ lf* and outperforms them by up to 1.53 points in R@1 *w/o lf* and 0.93 overall scores considering both settings. Ablations also verify the contributions of the dual encoder structure and the two-stage learning process. Further analyses suggest that both the target encoder and frame encoder can effectively capture the target-target, and frame-frame relationships.

Our contributions can be summarized as follow:

1. We propose COFFTEA, a dual encoder architecture with coarse-to-fine contrastive objectives. COFFTEA can effectively model frame definitions with learnable frame encoders and efficiently select the corresponding frame from over 1,000 frames.

2. We conclude the trade-off between *w/ lf* and *w/o lf*, and emphasize the performance *w/o lf* in the evaluation of frame identification. With two-stage learning, COFFTEA achieves an overall score improvement of 0.93 on frame identification.

3. Further in-depth analysis demonstrates that our frame and target encoders can better model the alignment between frame and target, as well as the relationships between target and target, frame and frame.

## 2 Task Formulation

For a sentence $S = w_1, \cdots, w_n$ with a target span $t = w_{t_s}, \cdots, w_{t_e}$, frame identification is to select the most appropriate frame $f \in \mathcal{F}$ for $t$, where $\mathcal{F}$ denotes all frames in FrameNet.

As FrameNet (Baker et al., 2007) provides each frame with an associated set of lexical units (LUs), we can obtain the LU of target $t$ by utilizing its lemma and part-of-speech (POS) in the form lemma.POS (e.g., ***get.v***) and apply *lf* to retrieve the lexicon filter candidate frames $\mathcal{F}_t$ of $t$. Then we focus on selecting $f \in \mathcal{F}_t$.

## 3 Methodology

To model the alignment between frames and targets efficiently and effectively, in this paper, we propose COFFTEA, a dual encoder architecture with two-stage contrastive objectives. We compare COFFTEA with other baselines modeling frame definitions (§ 3.1). Specifically, to achieve better represent learning of definitions, the frame encoder of COFFTEA is learnable during training. Moreover, in order to make COFFTEA gradually learn to differentiate frames with varying degrees of similarity. We introduce two-stage learning in §3.2.

### 3.1 Model Architecture

In Fig. 2, we compare the dual-encoder architecture of COFFTEA with three categories of previous meth-

Figure 2: Comparison between COFFTEA and other baselines. **(a)** and **(b)** both fine-tune a Lookup Table containing embeddings of frames with corss entropy loss. **(b)** uses a frozen frame encoder to initialize the table. **(c)** employs a single fused encoder to measure the probability of sentence and definition matching. For efficiency purposes, COFFTEA **(d)** also adopts a dual-encoder structure and leverages contrastive loss to unfreeze the frame encoder.

ods in frame identification: (a) no definition modeling (Hartmann et al., 2017; Zheng et al., 2022), (b) target encoder and frozen frame encoder (Su et al., 2021; Tamburini, 2022), and (c) single fused encoder (Jiang and Riloff, 2021). Limited by the number of frames in FrameNet, the dual encoder architecture with cross entropy loss (b) have to fix the parameters of the frame encoder while the single encoder architecture (c) have to train and infer under the *w/ lf* setting. Therefore, we still use two encoders to separately encode the sentence with targets and frame definition, and employ contrastive learning to make the frame encoder trainable.

We follow Su et al. (2021); Jiang and Riloff (2021); Tamburini (2022) to use PLMs (Devlin et al., 2019) as target encoder (§3.1.1) and frame encoder (§3.1.2).

### 3.1.1 Target Encoder

The target encoder converts input sentence $S$ with $n$ tokens $w_1, \ldots, w_n$ into contextualized representations $h_1, \ldots, h_n$. The contextualized representation $t$ of target span $w_{t_s}, \ldots, w_{t_e}$ ($1 \leq t_s \leq t_e \leq n$) is the maxpooling of $h_{t_s}, \ldots, h_{t_e}$ (Zheng et al., 2023).

$$h_1, \cdots, h_n = \text{Encoder}_t(w_1, \cdots, w_n) \quad (1)$$

$$t = \text{MaxPooling}(h_{t_s}, \cdots, h_{t_e}) \quad (2)$$

With the target encoder, we capture the contextual information of the target sentence and the sufficient modeling of context can be used to find out-of-vocabulary targets. Our goal is to not only ensure a close proximity between the representations of the target and the frame but also to ensure that the target representations in different sentences, which activate the same frame, exhibit a high degree of similarity.

### 3.1.2 Frame Encoder

Taking *Board_vehicle* as frame name and its corresponding definition *"A Traveller boards a Vehicle that they intend to use as a means of transportation either as a passenger or as a driver"* as def. Similarly, frame encoder converts $D =$ frame name|def, denoted as $w'_1, \ldots, w'_m$ into representations $h'_1, \ldots, h'_m$. We follow Reimers and Gurevych (2019) to regard meanpooling of all tokens as frame representations $f$.

$$h'_1, \cdots, h'_m = \text{Encoder}_f(w'_1, \cdots, w'_m) \quad (3)$$

$$f = \text{MeanPooling}(h'_1, \cdots, h'_m) \quad (4)$$

Through the frame encoder, we encode frame definitions to represent frames. Similar to the target

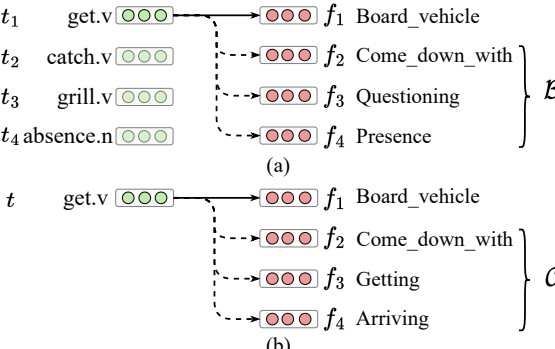

(a)

(b)

Figure 3: Coarse-to-Fine Learning of COFFTEA. In-batch learning **(a)** is first applied with coarse-grained in-batch negative examples $\mathcal{B}$. In-candidate learning **(b)** is then employed to differentiate gold frame with hard negatives $\mathcal{C}$.

encoder, the frame encoder not only focuses on aligning frames with targets but is also intended to reflect the semantic relationships between frames themselves, such as *Inheritance* (Su et al., 2021; Zheng et al., 2022).

## 3.2 Learning Process

In order to capture the alignment between frames and targets, we aim to minimize the distance between the target and the gold frame (positive pair) while maximizing the distance between the target and other frames (negative pairs) during the training process. However, encoding such a large number of frames, which exceeds 1000, and updating the encoders with gradients for all of them becomes infeasible. Therefore, we need to sample a subset of frames as negative pairs. As discussed earlier, both training *w/o lf* and *w/ lf* have their limitations. To address this, we propose a two-stage learning process in Fig. 3.

We first employ in-batch learning (§3.2.1), where we treat the gold frames of other targets in the same batch as negative examples. This allows us to learn coarse-grained frame semantic differences. Then we utilize in-candidate learning (§3.2.2), where we treat other frames in lexicon filter candidates as hard negatives. This step helps us refine the frame representations and capture fine-grained semantic distinctions

Specifically, we adopt cosine similarity $\cos(t, f) = \frac{t \cdot f}{\|t\| \cdot \|f\|}$ as distance metric and utilize contrastive learning to distinguish positive pairs from negative pairs. Let $\mathcal{L}_{\mathcal{B}}$ and $\mathcal{L}_{\mathcal{C}}$ denotes the contrastive loss of in-batch learning and in-candidate learning. The main difference between

| Datasets | exemplar | train | dev | test | $|\mathcal{F}|$ |
|----------|----------|-------|------|------|------|
| FN 1.5 | 153946 | 16621 | 2284 | 4428 | 1019 |
| FN 1.7 | 192431 | 19391 | 2272 | 6714 | 1221 |

Table 1: Statistics of FrameNet dataset

the two-stage learning lies in the construction of negative pairs $\mathcal{B}$ and $\mathcal{C}$.

$$\mathcal{L}_{\mathcal{B}} = -\log \frac{\exp\left(\cos(t, f^{+})/\tau\right)}{\sum_{f \in \mathcal{B} \cup \{f^{+}\}} \exp\left(\cos(t, f)/\tau\right)} \quad (5)$$

$$\mathcal{L}_{\mathcal{C}} = -\log \frac{\exp\left(\cos(t, f^{+})/\tau\right)}{\sum_{f \in \mathcal{C} \cup \{f^{+}\}} \exp\left(\cos(t, f)/\tau\right)} \quad (6)$$

### 3.2.1 In-batch Learning

Training directly *w/ lf* may cause the model to primarily focus on distinguishing the differences between the gold frame and the candidates. In order to broaden the model's horizons, we first conduct training *w/o lf*, allowing the model to learn to differentiate coarse-grained semantic differences. We follow Khosla et al. (2020) to construct in-batch negative examples $\mathcal{B}$.

### 3.2.2 In-candidate Learning

Through in-batch learning, our model can effectively differentiate some frames that are relatively easy to distinguish. However, we aim to further distinguish frames that are more likely to be confused with one another. Typically, when using *lf*, the other candidate frames for the same target can serve as negative examples $\mathcal{C}$.

However, the number of candidate frames for different targets can vary, and there are even cases where some targets only activate one frame in FrameNet. To address this issue, we pad the number of $\mathcal{C}$ for all targets to a fixed number (e.g., 15). This padding ensures that each target has a consistent number of candidate frames available for comparison during the training process.

We consider leveraging the semantic relationships between frames in FrameNet (§A.3) to expand $\mathcal{C}$. *Inheritance* represents the "is-a" relationship between frames. For example, *receiving* inherits from *getting*, then *getting* becomes the super frame and *receiving* becomes the sub frame. Other frames that also inherit from *getting*, such as *Commerce_buy* and *Taking*, are considered siblings of *receiving*. These siblings exhibit semantic similarities with *receiving*, but with subtle differences, making them suitable as hard negative examples.

| Models | FN 1.5 | | | | | FN 1.7 | | | | |
|---|---|---|---|---|---|---|---|---|---|---|
| | Acc | R@1 | R@3 | R@5 | Overall | Acc | R@1 | R@3 | R@5 | Overall |
| Hartmann et al. (2017)† | 87.63 | 77.49 | - | - | 82.25 | 83.00 | 76.10 | - | - | 79.40 |
| KGFI (2021)† | 92.13 | 85.63 | - | - | 88.76 | 92.40 | 85.81 | 91.59 | 92.88 | 88.98 |
| FIDO (2021)† | 91.30 | - | - | - | - | 92.10 | - | - | - | - |
| Tamburini (2022) | **92.57** | - | - | - | - | 92.33 | - | - | - | - |
| Lookup Table | 88.60 | 78.48 | 86.74 | 88.16 | 83.23 | 89.02 | 78.75 | 87.43 | 88.71 | 83.57 |
| + *init.* with Def | 90.40 | 83.56 | 88.25 | 89.32 | 86.85 | 90.30 | 85.17 | 88.93 | 89.87 | 87.66 |
| + $\mathcal{L}_{\mathcal{B}}$ | 91.06 | 86.52 | 91.73 | 92.68 | 88.73 | 90.94 | 87.13 | 92.30 | 93.21 | 89.00 |
| $\mathcal{L}_{\mathcal{B}}$ only† | 91.82 | 84.73 | 94.47 | 96.18 | 88.13 | 91.58 | 84.88 | 93.88 | 95.90 | 88.10 |
| $\mathcal{L}_{\mathcal{C}}$ only† | 92.50 | 76.51 | 87.35 | 91.01 | 83.75 | **92.70** | 78.33 | 88.93 | 91.64 | 84.91 |
| $\mathcal{L}_{\mathcal{B}}$ only | 90.94 | 86.77 | **95.14** | **96.79** | 88.81 | 91.36 | **87.38** | **95.29** | **96.77** | 89.33 |
| COFFTEA | 92.55 | **87.69** | 92.64 | 94.71 | **90.05** | 92.64 | 87.34 | 92.91 | 94.29 | **89.91** |

Table 2: Main results on FN1.5 and 1.7. **Acc** denotes accuracy **with** lexicon filtering and **R@**$k$ denotes recall at $k$ **without** lexicon filtering. **Overall** represents the harmonic mean of **Acc** and **R@1**, reflecting the comprehensive capability of models. Through dual encoders and two-stage learning, COFFTEA outperforms the previous methods (the topmost block) by at least 0.93 overall scores. Ablation study on dual encoders and two-stage learning is listed in the middle and bottom blocks (§ 4.4). † means the model does not trained with exemplar sentences.

We follow the order of candidate frames, sibling frames, and random frames to construct $\mathcal{C}$.

# 4 Experiment

Our experiments mainly cover three aspects:

- Main experimental results on frame identification: We compare the performance of COFFTEA with previous approaches both with lexicon filtering (*w/ lf*) and without lexicon filtering (*w/o lf*).

- Further ablation study: to demonstrate the contributions of the learnable frame encoder and Coarse-to-Fine learning to the performance of COFFTEA.

- In-depth representation analysis: We delve into a deeper analysis to investigate how the frame and target encoder of COFFTEA can better represent the alignment between frames and targets, as well as the relationships between frames and frames, targets and targets.

## 4.1 Datasets

We have introduced COFFTEA on two FrameNet datasets: FN 1.5 and FN 1.7. FN 1.7 is an extensive version of FN 1.5, with more annotated sentences and more defined frames. Following Swayamdipta et al. (2017); Su et al. (2021), we spilt the full text into train, dev and test set, and further use exemplars in FrameNet as pretraining data like Peng et al. (2018); Zheng et al. (2022); Tamburini (2022). Table 1 shows the statistics of two datasets.

## 4.2 Models

We mainly compare COFFTEA with representative previous methods from the different categories shown in Fig. 2: traditional representation learning based methods without definitions (Hartmann et al., 2017), dual encoder architectures with frozen frame encoders (KGFI (2021), Tamburini (2022)), and single fused encoder architectures (FIDO (2021)).

In order to conduct further ablation study and deeper analysis, we also implement two baselines represented as (a) and (b) in Fig 2: **Lookup Table** with random initialization and **Lookup Table initialized with definition representations**. The implementation details of previous methods, baselines, and COFFTEA can be found in §A.1.

## 4.3 Main Results

To compare our model with previous approaches, as well as our baselines. We report the accuracy (**Acc**) when lexicon filtering is applied, recall at k (**R@**$k$) when no lexicon filtering is used (indicating whether the gold frame is ranked in the top $k$ based on similarity), and an **Overall** score (harmonic average of **Acc** and **R@1**).

The upper block of Table 2 shows the performance of previous representative methods in frame identification. COFFTEA achieves a performance gain of 0.93 overall scores (88.98 v.s. 89.91), primarily due to a 1.53 increase in R@1 scores (85.81 v.s. 87.34) in the *w/o lf* setting. This improvement can be attributed to the use of a learnable frame

| Models | Predictions *w/o lf* from rank 2 to 5 (Top-1 is Becoming) | | | |
|---|---|---|---|---|
| $\mathcal{L}_\mathcal{B}$ only† | Transition_to_a_state | Undergo_transformation | Change_post_state | Undergo_change |
| $\mathcal{L}_\mathcal{C}$ only† | **Activity_start** | **Launch_process** | **Make_acquaintance** | **Meet_specifications** |

Table 3: Predictions in positions 2 to 5 of "The children and families who come to Pleasant Run are given the opportunity to **become** happy". **red** means the frame is irrelevant to the gold frame **Becoming**.

encoder and two-stage learning, which enhance the effectiveness of COFFTEA in modeling definitions. Compared to KGFI, COFFTEA achieves comparable Acc *w/ lf* (92.40 v.s. 92.64) and consistently performs better in R@$k$, which further demonstrates that through combining in-batch learning and in-candidate learning, COFFTEA is able to select the appropriate frame from both candidates (*w/ lf*) and all frames (*w/o lf*).

Moreover, it is worth noting that some methods (marked †) does not utilize exemplars as training data, while previous researches (Su et al., 2021; Zheng et al., 2022) believe that using exemplars for training improves accuracy. Further discussion about using exemplars can be found in §4.4.

In addition to representative methods in Table 2, there is a series of frame identification studies that solely focus on Acc *w/ lf*. We report the results of COFFTEA and more previous frame identification works in §B.2.

## 4.4 Ablation Study

**Frame Encoder** Can learnable frame encoder contribute to frame identification? As is shown in the middle block in Table 2, lookup table without definitions achieved relatively low overall scores (83.57). However, when we initialize the lookup table with the definitions encoded by the frozen encoder, the lookup table showed a significant improvement in R@1 *w/o lf*, with an increase of 6.42 points (78.75 v.s. 85.17). This indicates that even without adjusting the parameters, the frozen encoder of PLMs is capable of modeling the definitions to some extent.

To further explore its impact, we apply in-batch learning to the lookup table with definitions. This additional step results in a further improvement of 1.96 points in R@1 w/o lf (85.17 v.s. 87.13). This suggests that in-batch learning reduces the number of negative samples, decreases the learning difficulty, and ultimately leads to better performance.

Comparing COFFTEA to these lookup tables, we find that unfreezing the frame encoder allows for better modeling of fine-grained semantic differ-

ences, resulting in a significant increase of 1.70 points in accuracy *w/ lf* (90.94 v.s. 92.64). This demonstrates the effectiveness of incorporating an unfrozen frame encoder to enhance the modeling of subtle semantic distinctions and improve the performance of the frame identification task.

**Coarse-to-Fine Learning** The bottom block in Table 2 demonstrates the trade-off between in-batch learning ($\mathcal{L}_\mathcal{B}$ only) and in-candidate learning ($\mathcal{L}_\mathcal{C}$ only). $\mathcal{L}_\mathcal{B}$ **only** achieves the highest R@1 (84.88) and $\mathcal{L}_\mathcal{C}$ **only** yields the highest Acc (92.70). However, the former approach does not excel at distinguishing fine-grained frames (91.58 Acc), while the latter approach even performs worse than the lookup table on R@1 *w/o lf* (78.33 v.s. 78.75).

Table 3 gives a specific example for comparison. Both methods can correctly predict the frame corresponding to the target *become* in the given sentence. However, when looking at the predicted frames in positions 2 to 5, the frames predicted by in-batch learning show structural relationships with *Becoming*, while the frames predicted by in-candidate learning have no connection to *becoming*. This reflects that in-candidate learning cannot distinguishing frames sufficiently.

To address this issue, we train COFFTEA with $\mathcal{L}_\mathcal{B}$ on exemplars and then train COFFTEA with $\mathcal{L}_\mathcal{C}$ with train split, which is equivalent to performing coarse-grained pretraining on a large-scale corpus and then fine-tuning on a smaller dataset for domain adaptation at a finer granularity.

We also apply in-batch learning on both train and exemplar. Due to the lack of hard negatives, its performance on Acc *w/ lf* is lower than COFFTEA (91.36 v.s. 92.64).

**Exemplars** Using exemplars as additional training data is a common practice (Peng et al., 2018; Chen et al., 2021; Zheng et al., 2022). Since in-batch learning benefits from larger-scale data (84.88 v.s. 87.38 in Table 2), we chose to use the larger set, exemplars for the first stage of in-batch learning. However, this makes our results not directly comparable to those of previous methods (Su

| Models | w/ exem. | w/o exem. | Δ |
|---|---|---|---|
| Tamburini (2022) | 91.42 | 92.33 | -0.91 |
| Lookup Table | 89.02 | 89.41 | -0.39 |
| + *init.* with Def | 90.40 | 90.66 | -0.26 |
| + $\mathcal{L}_\mathcal{B}$ | 91.06 | 91.23 | -0.17 |
| $\mathcal{L}_\mathcal{B}$ only | 91.36 | 91.58 | -0.22 |
| COFFTEA | **92.55** | **92.24** | **0.31** |

Table 4: Training with extra exemplars does not increase the model performance. COFFTEA can effectively utilize exemplars via two-stage learning.

| Models | Acc | Acc w/ [MASK] | Δ |
|---|---|---|---|
| Lookup Table | 89.02 | 78.11 | 10.92 |
| + *init.* with Def | 90.30 | 77.58 | 12.72 |
| + $\mathcal{L}_\mathcal{B}$ | 90.94 | 77.91 | 13.03 |
| COFFTEA | **92.64** | **85.82** | **6.82** |

Table 5: Masking target words can cause the decrease of accuracy. The extent of the decrease, Δ, reflects whether the model understands the context information of the target.

| Models | R@1 | 10 target exemplars | | |
|---|---|---|---|---|
| | | R@1 | R@3 | R@5 |
| Lookup Table | 78.75 | 63.96 | 75.54 | 77.27 |
| + *init.* with Def | 85.17 | 81.71 | 89.51 | 90.33 |
| + $\mathcal{L}_\mathcal{B}$ | 87.13 | 83.04 | 92.24 | 93.40 |
| COFFTEA | **87.34** | **85.58** | **93.07** | **94.89** |

Table 6: Representing frames with definitions and ten target exemplars. The target encoder of COFFTEA can get similar and consistent representations of targets related to the same frame.

| Models | Average $\Delta\alpha/\alpha$ |
|---|---|
| Lookup Table | 0.158 |
| + *init.* with Def | 0.012 |
| + $\mathcal{L}_\mathcal{B}$ | 0.070 |
| $\mathcal{L}_\mathcal{B}$ only† | 56.290 |
| $\mathcal{L}_\mathcal{C}$ only† | 6.494 |
| COFFTEA | **121.816** |
| COFFTEA w/o. sibling | 34.907 |

Table 7: The learnable frame encoder better models the structured information between frames. The perception of structure in COFFTEA comes from the first stage of in-batch learning and the utilization of siblings as hard negatives.

et al., 2021) that only use the training set. However, more training data does not necessarily lead to improved performance. As shown in Table 4, the performance of Tamburini (2022) and our lookup table baselines decreases after using exemplars, possibly due to a domain mismatch between the exemplars and the training set. However, our two-stage approach effectively utilizes both datasets to get an improvement of 0.31 Acc, providing a more appropriate way to utilize exemplars for future work.

### 4.5 Further Analysis

**Target-Frame Alignments** The results in Table 2 demonstrate that COFFTEA effectively models the alignment between targets and frames. However, we still want to explore whether the model's ability to predict a frame is due to its accurate modeling of the target's contextual information in the sentence or if there are other shortcuts involved. Moreover, if COFFTEA can effectively model the context of the target, it can also be used to discover out-of-vocabulary targets. To investigate this, we replace the target in the sentence with the [MASK] token and compute the similarity between the masked target representation and frames. Table 5 shows that when the target is masked, the accuracy of the model decreases across the board. COFFTEA achieves the smallest decrease in performance (Δ), indicating the capability of our dual-encoder structure to effectively model the alignment between target context and frames. This capability also indicates that COFFTEA can explore out-of-vocabulary targets.

**Target-Target Relations** The target encoder ensure a close proximity between targets and their corresponding frames. However, we believe that different sentences the target representations in different sentences, which activate the same frame, are supposed to exhibit a high degree of similarity. We select ten exemplars for each frame and replace frame representations with the mean representation (centroid) of these exemplars. In Table 6, we compare two frame representations: directly encoding the definition and encoding ten exemplars. COFFTEA achieves consistent and similar performance in both settings (87.34 and 85.58), reflecting the similarity between the target representations of the same frame and the final frame representation, as captured by our target encoder.

**Frame-Frame Relations** Due to the semantic relations defined by FrameNet between frames (such as *inheritance*, *using*, *subframe*, etc.), frames are not isolated but rather interconnected with each other, forming structural relationships. For exam-

ple, if *receiving* inherits from *getting*, their representations should be similar. Frame representations should reflect the structural relationships. To evaluate the degree of similarity between a subframe $f_{sub}$ and its superframe $f_{sup}$, we define a metric,

$$\alpha = \frac{1}{|\mathcal{F}|} \cdot \sum_{f \in \mathcal{F}} \cos(f_{sub}, f) \qquad (7)$$

$$\Delta\alpha = \cos(f_{sub}, f_{sup}) - \alpha \qquad (8)$$

$\frac{\Delta\alpha}{\alpha}$ is a normalized evaluation reflecting how a subframe is close to its superframe compared to other frames. We report the average $\frac{\Delta\alpha}{\alpha}$ ($\alpha > 0$) of *inheritance* pairs in Table 7.

The baselines (the upper block) without a learnable frame encoder struggle to model the relationships between frames effectively. The distance between subframes and superframes is almost the same as that between other frames. However, using a learnable frame encoder greatly improves the similarity between subframes and superframes.

In-candidate learning, if used alone, tends to overly focus on differentiating frames with similar semantics, which actually decreases the similarity between them (6.494 v.s. 56.290). On the other hand, COFFTEA effectively models the similarity between associated frames. This is mainly because we use two-stage learning and treat siblings as hard negative examples during training (34.907 v.s. 121.816), which indirectly enables the model to learn the connection between subframes and superframes.

## 5 Related Work

**Frame Identification**   Prior research in frame identification has primarily focused on modeling the similarity between targets and frames or assessing the matching score between them.

In terms of modeling similarity, Hartmann et al. (2017); Peng et al. (2018); Zheng et al. (2022) leverage target encoder with a trainable frame representation table, while ignoring the valuable contextual frame information provided by FrameNet. To remedy this deficiency, Su et al. (2021); Tamburini (2022) both propose to incorporate frame definitions into frame encoder. Despite these efforts, the number of frames and the optimizing objective make it infeasible to update frame representation during training, leading to a compromise of freezing it. Consequently, they all trap in the dilemma to obtain optimal frame representation. As for assessing the matching score, Jiang and Riloff (2021)

propose a fused encoder to evaluate semantic coherence between sentence and frame definition. Nevertheless, it just fails to evaluate the performance of *w/o lf* scenario.

In addition, the aforementioned studies tend to adopt a one-sided relationship between targets and frames. They only train model on either *w/ lf* or *w/o lf*, which can not catch an overall horizon in distinguishing frames. Moreover, only Su et al. (2021) release performance under *w/o lf* scenario, which is essential for evaluating the full capabilities of the model.

Some other work like Chanin (2023) treat frame semantic parsing as a sequence-to-sequence text generation task, but its still trap in training under *w/ lf* and can not catch an optimal relationships.

**Metric Learning**   Metric learning is a fundamental approach in machine learning that seeks to learn a distance metric or similarity measure that accurately captures the inherent relationships between data points. One particular technique that has gained prominence in this area is Contrastive Learning, which has been extensively applied to a variety of tasks such as text classification (Gao et al., 2021), image classification (He et al., 2020).

The optimized feature space retrieved from Contrastive Learning objective can benefit on learning better representations of data and capturing inner relationship. For frame identification, the primary goal is to obtain a well-aligned target-to-frame space. However, existing researches only focus on using Cross Entropy to directly modeling correlation to All Frames, which presents challenges in capturing complex relationships. Alternatively, Contrastive Learning can be leveraged to optimize and incrementally construct the feature space of frames and targets.

## 6 Conclusion

We propose COFFTEA, and its dual encoder can effectively model the definitions with learnable frame encoder and efficiently calculate the similarity scores between a target and all frames. We also conclude that there exists a trade-off between training *w/ lf* and *w/o lf*, and achieves a comprehensive performance with a two stage learning process. Detailed analysis of the experiments further demonstrates that our two encoders are not limited to their alignment but can also discover other connections between frame and frame, as well as target and target.

## Limitations

The limitations of our work include:

1. We only used cosine similarity as a distance metric and did not explore other methods of metric learning. Investigating alternative distance metrics could potentially enhance the performance of our model.

2. Currently, the FrameNet dataset does not provide test data for out-of-vocabulary words. Therefore, our exploration of out-of-vocabulary words is limited to preliminary experiments (Table 5). Further investigation with a dedicated dataset containing out-of-vocabulary test data would be valuable.

3. We only applied the dual-encoder architecture to frame identification and did not explore its application to frame semantic role labeling. Extending our approach to frame semantic role labeling would provide insights into its effectiveness and generalizability beyond frame identification.

## Acknowledgements

We thank all reviewers for their great efforts. This paper is supported by the National Science Foundation of China under Grant No.61936012.

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

## A  Experiment Details

### A.1  Baseline Models

**Hartmann et al. (2017)**: a simple classifier based on the representation of entire sentence and especially dependents of target.

**Peng et al. (2018)**: a joint model to learn semantic parsers from disjoint corpora with a latent variable formulation.

**Chen et al. (2021)**: a joint model of multi-decoder and hierarchical pointer network for frame semantic parsing.

**FIDO (2021)**: a fused model assessing semantics coherence between sentence and frame definition.

**KGFI (2021)**: a dual encoders model for frame identification aligning similarity of targets and pre-computed frame representation.

**Tamburini (2022)**: similar to Su et al. (2021) with more reliable graph technique.

**KID (2022)**: a double graph based model for frame semantic paring.

**AGED (2023)**: a query-based framework for Frame Semantic Role Labeling under frame definition.

**Chanin (2023)**: a T5-based model treating frame semantic parsing as sequence-to-sequence text generation task.

**Lookup Table**: a baseline model aligning target encoder with trainable one-hot mapping frame representation.

**Lookup Table init. with definition**: a baseline model aligning target encoder with trainable BERT-initialized frame representation.

**Lookup Table init. with definition and $\mathcal{L}_\mathcal{B}$**: using Contrastive Loss as optimize objective instead of Cross Entropy.

### A.2  Hyper-parameter Setting

For replicability of our work, we list the Hyper-parameter setting of all our baselines and COFFTEA in Table 8.

### A.3  Frame Relation

Given a frame $f$, to get its sibling frames $\mathcal{F}_{sib}$, we utilize the frame relation *Inheritance* and consider $f$ as the superior or subordinate frame of *Inheritance* to retrieve its subordinate or superior frames, and then get the sibling of $f$. Table 9 shows the *Inheritance* relation we used.

| Hyper-parameter | Value |
|---|---|
| Epochs | 20 |
| Learning Rate | 2e-5 |
| Optimizer | AdamW |
| Batch Size of $\mathcal{L}_\mathcal{B}$ | 32 |
| Gradient Accumulation of $\mathcal{L}_\mathcal{B}$ | 4 |
| Temperature of $\mathcal{L}_\mathcal{B}$ | 0.07 |
| Batch Size of $\mathcal{L}_\mathcal{C}$ | 6 |
| Gradient Accumulation of $\mathcal{L}_\mathcal{C}$ | 3 |
| Candidate Frame Number of $\mathcal{L}_\mathcal{C}$ | 15 |
| Temperature of $\mathcal{L}_\mathcal{C}$ | 1 |
| Dual Encoders | BERT-base |

Table 8: Hyper-parameter settings

| Sup frame | Sub frame |
|---|---|
| Getting | Receiving
Amassing
Commerce_buy
Commerce_collect
Taking |
| Receiving | Borrowing |
| Transition_to_a_state | Becoming
Undergo_change |
| Undergo_change | Undergo_transformation |
| Becoming | Transition_to_a_quality |

Table 9: A part of pairs of *Inheritance* relation, see more at https://framenet.icsi.berkeley.edu/fndrupal/.

## B  Supplementary Experiment Results

### B.1  Statistical Results of COFFTEA

To verify that our experimental results were not accidental, we train COFFTEA with five different random seeds for each dataset. Compare with the strongest model (Su et al., 2021), we can get better performance consistently. Table 10 shows the average performances with its deviation.

| Dataset | Models | Avg. $\pm$ Dev. | |
|---|---|---|---|
| | | w/ lf | w/o lf |
| FN1.5 | KGFI | 92.40 | 84.41 |
| | COFFTEA | **92.51** $\pm$ 0.14 | **87.26** $\pm$ 0.10 |
| FN1.7 | KGFI | 92.13 | 85.63 |
| | COFFTEA | **92.50** $\pm$ 0.05 | **87.29** $\pm$ 0.41 |

Table 10: Statistical results of multiple runs on FN1.7. We train our models in with five different random seeds and report the average performance with deviation.

### B.2  Comparison with More Methods

As most previous work only focus on performance in the *w/ lf* setting, we conduct a comprehensive

comparison on both **All** frames and **Amb** frames, where **Amb** is a subset of **All** whose target is polysemous or can evkoe multiple frames. And we find that Tamburini (2022) just use a smaller test set on both FN1.5 and FN1.7. So our COFFTEA achieves competitive performance compared to the strongest models.

| Models | FN 1.5 | | FN 1.7 | |
|---|---|---|---|---|
| | **All** | **Amb** | **All** | **Amb** |
| Hartmann et al. † | 87.63 | 73.80 | 83.00 | 71.70 |
| Peng et al. | 90.00 | 78.00 | 89.10 | 77.50 |
| Chen et al. | 90.50 | 79.10 | - | - |
| FIDO† | 91.30 | 81.00 | 92.10 | 83.80 |
| KGFI† | 92.13 | 82.34 | 92.40 | 84.41 |
| AGED | 91.63 | 81.57 | - | - |
| KID | 91.70 | 81.72 | 91.70 | 83.03 |
| Tamburini | **92.57** | 83.58 | 92.33 | 84.22 |
| Chanin | - | - | 89.00 | 77.50 |
| COFFTEA | 92.55 | **83.61** | 92.64 | **84.95** |

Table 11: Frame identification accuracy with lexicon filtering on FrameNet test dataset, 'All' and 'Amb' denote testing on test data and ambiguous data respectively. † means the model does not trained with exemplar sentences.