# OpenReview forum: "Coarse-to-Fine Dual Encoders are Better Frame Identification Learners"
_EMNLP/2023/Conference — EMNLP 2023 Findings_

### Official Review · Reviewer_nLkN · 2023-08-03

**Soundness:** 3

**Excitement:**

4: Strong: This paper deepens the understanding of some phenomenon or lowers the barriers to an existing research direction.

**Paper Topic And Main Contributions:**

- Proposes a dual encoder model  for frame identification using contrastive learning.
- It adopts dual encoder structure for efficiency.
- It introduces a two-stage coarse-to-fine learning procedure to handle trade-off between training with and without lexicon filtering.

**Reasons To Accept:**

- The paper is well written and easy to follow.
- Dual encoder structure allows efficiently computing similarity between target and all frames.
- The proposed coarse-to-fine learning addresses limitations of only training with or without lexicon filtering.
- The method achieves strong performance on frame identification benchmarks, especially without lexicon filtering.

**Reasons To Reject:**

- The overall dual-encoder framework and contrastive learning are lack of novelty. There is no new methodology proposed and the framework described is basically incremental.
- The paper claims the ability of addressing out-of-vocabulary targets, but no experimental analysis is provided.

**Reproducibility:**

4: Could mostly reproduce the results, but there may be some variation because of sample variance or minor variations in their interpretation of the protocol or method.

**Reviewer Confidence:**

3: Pretty sure, but there's a chance I missed something. Although I have a good feel for this area in general, I did not carefully check the paper's details, e.g., the math, experimental design, or novelty.

---

> ### Author Rebuttal · Authors · 2023-08-28
>
> We really appreciate your effort in reviewing our paper, and thank you for your helpful comments. We are glad to see that you recognize 1) our model efficiently align target with all frames 2) our method well-motivated and effective. Now we address your questions below:
>
> Q1: The overall dual-encoder framework and contrastive learning are lack of novelty. There is no new methodology proposed and the framework described is basically incremental.
>
> A1: In this paper, **our innovation depends on our focus on and analysis of the two overlooked issues in Frame Identification  - lexicon filtering and insufficient learning**. Ideally, we can use contrastive learning to perfectly align target with all frames and dual-encoder framework to sufficiently promote effective learning of frames.
>
> At the same time, we precisely design methods to prove that our two-stage learning is mutually reinforcing, rather than simply piling up. Our three parts in Section 4.5 respectively demonstrate that:
> 1) we learn **good alignments between targets and frames**, where we get SOTA performance on two benchmarks and both scenarios.
> 2) we **obtain better target representation which fully understand the contextual context**. In Table 6, when using the learned exemplars' target representation to recall frames, **COFFTEA get 85.58, the closest to w/o. lf 87.34**.
> 3)  our **frame representations rich in frame relationships**, which enable us to maximize the differentiation between subframes with inheritance relationships relative to superframes and other frames, **with highest scores of 121.816**.
>
> Q2: The paper claims the ability of addressing out-of-vocabulary targets, but no experimental analysis is provided.
>
> A2: As our limitation states, currently, the FrameNet dataset does not provide test data for out-of-vocabulary words. Therefore, our exploration of out-of vocabulary words is limited to preliminary experiments. **Based on the assumption of word distribution, words with the same semantics often appear in the same context**. Therefore, **by masking the corresponding target, we can simulate our model's ability to match the corresponding frame when out-of-vocabulary targets appear**. In Table 5, when target is masking, **our COFFTEA declines least by 6.82% compared to the baselines decrease above 10%**, indicating our strongest potential to handle out-of-vocabulary targets.

---

### Official Review · Reviewer_9bh3 · 2023-08-10

**Soundness:** 3

**Excitement:**

3: Ambivalent: It has merits (e.g., it reports state-of-the-art results, the idea is nice), but there are key weaknesses (e.g., it describes incremental work), and it can significantly benefit from another round of revision. However, I won't object to accepting it if my co-reviewers champion it.

**Paper Topic And Main Contributions:**

This paper proposes a COFFTEA, a Coarse-to-Fine Frame and Target Encoders Architecture. By contrastive learning  and dual encoders, COFFTEA efficiently and effectively models the alignment between frames and targets. With a coarse-to-fine curriculum learning procedure, the proposed method gradually learns to differentiate frames with varying degrees of similarity.

**Reasons To Accept:**

I am not the expert in the topic, but I think the paper is very easy to read.
The motivation is clear, and the proposed solution shows reasonable for addressing the problem.


**Reasons To Reject:**

Some typos should be revised.

**Reproducibility:**

3: Could reproduce the results with some difficulty. The settings of parameters are underspecified or subjectively determined; the training/evaluation data are not widely available.

**Reviewer Confidence:**

1: Not my area, or paper was hard for me to understand. My evaluation is just an educated guess.

---

> ### Author Rebuttal · Authors · 2023-08-29
>
> We really appreciate your effort in reviewing our paper, and we are glad to see that you recognize our writing and motivation. We will proofread and revise our typos carefully.

---

### Official Review · Reviewer_wf6g · 2023-08-10

**Soundness:** 4

**Excitement:**

4: Strong: This paper deepens the understanding of some phenomenon or lowers the barriers to an existing research direction.

**Paper Topic And Main Contributions:**

This paper seeks to enhance the correspondence between linguistic words and video frames in the context of frame identification. Recognizing the ambivalent nature of the commonly employed lexicon filtering (LF) strategy, this study introduces a two-stage learning framework that initially incorporates LF and subsequently omits it. Furthermore, a dual encoder structure is adopted within this framework. Through comprehensive experimentation, the effectiveness of the proposed components within the framework is validated.

**Reasons To Accept:**

1. This paper provides a clear motivation by highlighting the advantages and disadvantages of the widely-used lexicon filtering (LF) strategy. Consequently, a corresponding method is proposed to further enhance learning with LF.

2. Each component within the proposed framework is both effective and straightforward, offering promising prospects for replication with the provided code.

3. The paper is well-written and predominantly easy to comprehend.

**Reasons To Reject:**

There are only a few minor issues that need to be addressed:
The proposed method lacks clear validation of the mutual benefits resulting from the two-stage training approach, particularly in terms of specific metrics such as ACC. For instance, on the FN1.7 dataset, the overall framework (COFFTEA) demonstrates slightly lower performance compared to $L_{C}$ only†. It would be helpful to provide further clarification on this matter.

**Reproducibility:**

4: Could mostly reproduce the results, but there may be some variation because of sample variance or minor variations in their interpretation of the protocol or method.

**Reviewer Confidence:**

3: Pretty sure, but there's a chance I missed something. Although I have a good feel for this area in general, I did not carefully check the paper's details, e.g., the math, experimental design, or novelty.

---

> ### Author Rebuttal · Authors · 2023-08-28
>
> We really appreciate your effort in reviewing our paper, and thank you for your helpful comments. We are glad to see that you recognize the rationality of our motivation and the effectiveness of our method. Now we address your questions below:
>
> Q1: The proposed method lacks clear validation of the mutual benefits resulting from the two-stage training approach, particularly in terms of specific metrics such as ACC.
>
> A1: We really appreciate your interest in the mutual benefits of our two-stage learning and would like to address your concerns. In Table 1, we list the results of our each stage, where **$L_B$ performs better on w/o. LF (84.88 v.s. 78.33) while  $L_C$ performs better on w. LF (92.70 v.s. 91.58)**. To clarify this phenomenon, we give an example of the prediction result in Table 3, the frames predicted by **$L_B$ show structural relationships with golden frame, while $L_C$ just has no connection**. This indicates that **no matter which stage is used alone, the alignment learned is insufficient**, but our COFFTEA realizes to combine the advantages and achieves the best overall performance with 0.93 improvement.
>
> In addition to using quantitative indicators like ACC, we also designed a set of experiments to demonstrate the mutual benefits of COFFTEA in Section 4.5. One of them demonstrates that **our model can better learn the relation between frames**. We define a metric refer to Eq 7 to measure the degree of similarly between a subframe and its superframe compared with other frames. In Table 7, we show that the baselines without a trainable frame encoder struggle to distinguish all frames, with a degree about 0.100, while **COFFTEA can surprisingly simulate the role of two stages and get a score of 121.816**, indicating COFFTEA effectively models the similarity between associated frames.

---

### Meta-Review · Area_Chair_neGi · 2023-09-15

**Recommendation:** 3

**Metareview:**

In this work, the authors tackle the longstanding CL task of frame identification (matching of words in context with semantic frames they induce) and propose an approach based on reshaping of a representation space to relfect the topology of frame-based similarity.

The approach, in principle very similar to approaches that convert language models into sentence encoders, consists of: (1) dual encoding (separate encoding of the context of the target word and the sentential example(s) that instantiate the frame); (2) contrastive learning (contrastive loss effectively reshapes the encoder so that the proximity of vectors of sentential instances reflects their alignment in terms of the semantic frame they correspond to, i.e., which they induce); and (3) coarse-to-fine curriculum learning in which the difficulty of negative examples is gradually increased (i.e., one starts with randomly selected, and likely very different semantic frames as negatives in early stages of training and then progresses to use semantically closer frames as negatives in later stages). The authors show empirically that all three components are important for the reported performance gains.

All reviewers agree that the work is sound and identify no errors in evaluation protocols. The proposed approach is meaningful and yields gains, although the gains are not particularly large. The reviewers -- and I fully support this view are somewhat more skeptical of the methodological novelty: all three components, namely, the bi-encoding, contrastive learning, and curriculum with increasingly difficult negatives are very widely and commonly used techniques, widely used in sentence-level classification and ranking tasks. From this point of view, the work can be seen as an application of this well-known framework to the task of frame identification (where instances are also sentences or quasi-sentences).

Overall, I find this work to be sound and valuable for the CL community and the narrower community that focuses on frame identification, but not particularly exciting for the wider NLP audience.

---

### Decision · Program_Chairs · 2023-10-07

**Decision:**

Accept-Findings

**Comment:**

In this work, the authors tackle the longstanding CL task of frame identification (matching of words in context with semantic frames they induce) and propose an approach based on reshaping of a representation space to relfect the topology of frame-based similarity.

The approach, in principle very similar to approaches that convert language models into sentence encoders, consists of: (1) dual encoding (separate encoding of the context of the target word and the sentential example(s) that instantiate the frame); (2) contrastive learning (contrastive loss effectively reshapes the encoder so that the proximity of vectors of sentential instances reflects their alignment in terms of the semantic frame they correspond to, i.e., which they induce); and (3) coarse-to-fine curriculum learning in which the difficulty of negative examples is gradually increased (i.e., one starts with randomly selected, and likely very different semantic frames as negatives in early stages of training and then progresses to use semantically closer frames as negatives in later stages). The authors show empirically that all three components are important for the reported performance gains.

All reviewers agree that the work is sound and identify no errors in evaluation protocols. The proposed approach is meaningful and yields gains, although the gains are not particularly large. The reviewers -- and I fully support this view are somewhat more skeptical of the methodological novelty: all three components, namely, the bi-encoding, contrastive learning, and curriculum with increasingly difficult negatives are very widely and commonly used techniques, widely used in sentence-level classification and ranking tasks. From this point of view, the work can be seen as an application of this well-known framework to the task of frame identification (where instances are also sentences or quasi-sentences).

Overall, I find this work to be sound and valuable for the CL community and the narrower community that focuses on frame identification, but not particularly exciting for the wider NLP audience.